# Remote Sensing Based Yield Estimation of Rice (*Oryza Sativa* L.) Using Gradient Boosted Regression in India

Ponraj Arumugam [1,*], Abel Chemura [1], Bernhard Schauberger [1] and Christoph Gornott [1,2]

1   Potsdam Institute for Climate Impact Research (PIK), Member of the Leibniz Association,
    14473 Potsdam, Germany; chemura@pik-potsdam.de (A.C.); schauber@pik-potsdam.de (B.S.);
    gornott@pik-potsdam.de (C.G.)
2   Agroecosystem Analysis and Modelling, Faculty of Organic Agricultural Sciences, University of Kassel,
    37213 Kassel, Germany
*   Correspondence: ponraj@pik-potsdam.de

**Abstract:** Accurate and spatially explicit yield information is required to ensure farmers' income and food security at local and national levels. Current approaches based on crop cutting experiments are expensive and usually too late for timely income stabilization measures like crop insurances. We, therefore, utilized a Gradient Boosted Regression (GBR), a machine learning technique, to estimate rice yields at ~500 m spatial resolution for rice-producing areas in India with potential application for near real-time estimates. We used resampled intermediate resolution (~5 km) images of the Moderate Resolution Imaging Spectroradiometer (MODIS) Leaf Area Index (LAI) and observed yields at the district level in India for calibrating GBR models. These GBRs were then used to downscale district yields to 500 m resolution. Downscaled yields were re-aggregated for validation against out-of-sample district yields not used for model training and an additional independent data set of block-level (below district-level) yields. Our downscaled and re-aggregated yields agree well with reported district-level observations from 2003 to 2015 ($r = 0.85$ & MAE = 0.15 t/ha). The model performance improved further when estimating separate models for different rice cropping densities (up to $r = 0.93$). An additional out-of-sample validation for the years 2016 and 2017, proved successful with $r = 0.84$ and $r = 0.77$, respectively. Simulated yield accuracy was higher in water-limited, rainfed agricultural systems. We conclude that this downscaling approach of rice yield estimation using GBR is feasible across India and may complement current approaches for timely rice yield estimation required by insurance companies and government agencies.

**Keywords:** yield estimation; high resolution; Remote Sensing; MODIS; Leaf Area Index (LAI); machine learning; Gradient Boosted Regression (GBR); India

## 1. Introduction

Accurate information on crop acreage, yield, and production is required for loss determination in crop insurance schemes and adaptation strategies' development. This need is even more urgent in countries like India, where the agricultural sector is a livelihood for millions of farmers, with changes having far-reaching impacts on food security and the economy. In India, 58% of the population work in the agricultural sector, and more than 75% of the country's farmers are smallholders [1,2]. One of India's major staple crops is rice; it is cultivated on about 150 million hectares, producing 132 million metric tons, over a quarter of the global rice production [3].

In the last decade, rapid advances in remote sensing (RS) and machine learning (ML) techniques provided cost-effective and comprehensive solutions for agro-environmental monitoring and decision making [4]. In India, most large-scale yield estimation studies have used remote sensing vegetation indices (i.e., The normalized difference vegetation index—NDVI, the enhanced vegetation index—EVI, and Soil-adjusted vegetation index—SAVI) and biophysical parameters (i.e., Leaf Area Index—LAI, the fraction of Absorbed

Photosynthetically Active Radiation—fAPAR, and chlorophyll content) to estimate yields for various crops [5–7]. Forecasting Agricultural output using Space, Agro-meteorology, and Land-based observations (FASAL) is a program in India to predict crop yields at the national scale. Under the FASAL program, crop area and yields are predicted using various methods separately, such as weather-yield models, crop simulation models, and remote sensing-driven statistical models [8–12] but limited to ML approaches. Few studies have merged remote sensing with machine learning approaches to estimating crop yield on a large scale in various parts of the globe [13–15]. Using recent advancements in data and methods for a large-scale yield assessment in a country like India is needed in many applications like informing the loss determination for crop insurances or suggesting management strategies for agricultural extensions [16].

The aim of our study was to implement a Gradient Boosted Regression (GBR), a machine learning approach, to estimate rice yield from MODIS LAI at 500 m spatial resolution for rice-producing areas in India. GBR has been used widely because of its high accuracy, fast training, prediction time, and small memory footprint in various applications [17]. The LAI is broadly used for crop growth monitoring and estimate yields, and it is considered an outcome of several factors, such as weather, water availability, and crop development. Various biophysical crop model simulations use LAI through data assimilation techniques [18–20] to improve the yield estimation accuracy. It has been established that intra-seasonal phenological patterns, including crop stress, can be captured using remotely sensed time-series data [21].

In this study, we focused on the following objectives; (a) Developing an efficient framework (downscaling approach) to establish a yield model with publicly available data. (b) Testing an ML approach (GBR) for accurate yield model development at a national scale. (c) Assessing the efficiency of the downscaling approach. (d) Identifying the scalability of this approach for agriculture applications at the regional scale. To achieve these objectives, we followed four steps in our study. First, we upscaled (to 5 km) available remote sensing data (500 m) and reported yield data at the district level. Second, we developed GBR models from upscaled rice yields and LAI information, using data between 2003 to 2015. Third, we applied the developed GBR models to estimate rice yields at 500 m using MODIS LAI inputs at the original spatial resolution. Finally, we validated our approach with two independent data sets, once for 2016 and 2017 district yields and independent block-level data in Maharashtra from 2003–2015, by aggregating 500 m spatial resolution yields to the particular administrative units.

## 2. Materials and Methods

### 2.1. Study Area

In this study, remote sensing data and Gradient Boosted Regression (GBR) were used to estimate rice yield for the Kharif season from 2003 to 2017 at 500 m spatial resolution in India. India consists of 36 states, 684 districts, and 5969 sub-districts or blocks (2015 census). Based on the availability of Kharif season observed yields and significance in rice-growing areas, 18 states (~62% of all India rice areas) were considered here. We developed and validated separate models for each state. Figure 1 shows the study area with state administration boundaries and a rice crop mask.

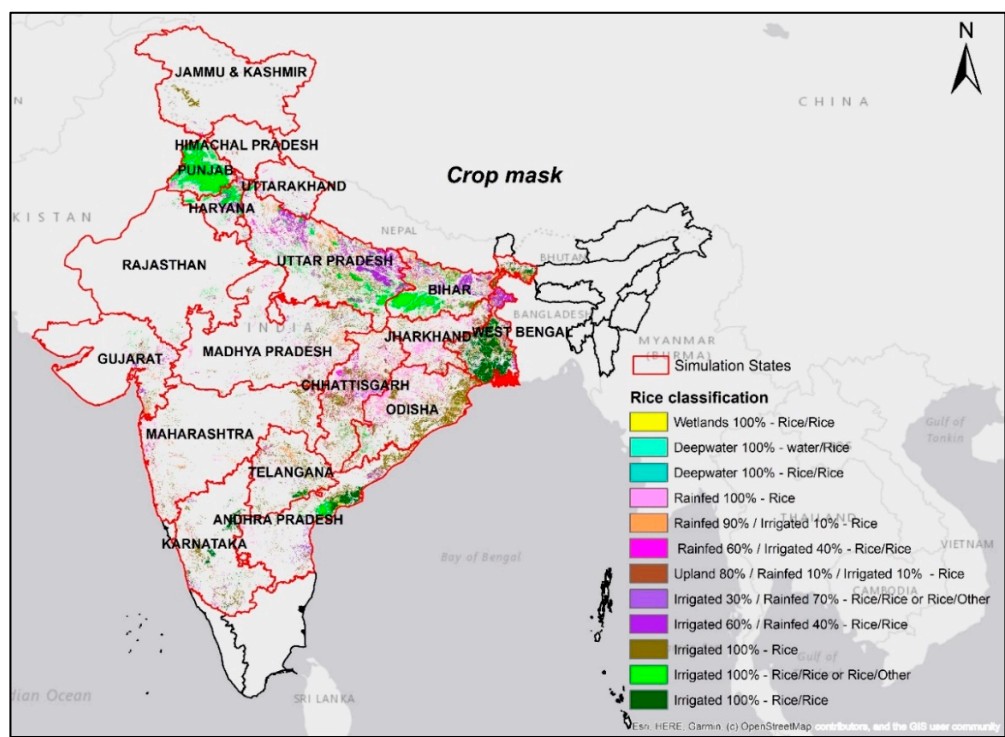

**Figure 1.** The rice classification map with study area state boundaries.

## 2.2. Methodology Overview

For our study, we used the MODIS Leaf Area Index (LAI) values at 500 m spatial resolution, which is available at 8 days temporal interval, from Julian Day 161 to 361, and reported rice yields at the district-level to predict rice yield at 500 m resolution between 2003 and 2015. We resampled both LAI and yield values to an intermediate spatial resolution of 5 km for developing gradient boosted regression models to estimate yields. LAI values within each 5 km × 5 km were averaged without area weighting or masking. The district-level yields were spread on a grid with 5 km spatial resolution by assigning the same yield level to all grid cells within each district based on majority coverage (i.e., if any cell intersects multiple districts, it will be populated with the yield observation from the district with the largest overlap). This 5 km spatial resolution was selected for multiple reasons: to achieve a sufficient number of training pixels per district (~30), capture the variance of LAI across grid levels, achieve more accurate models, and limit computation time. The average rice area per district in India is approximately 75,000 ha, with a total rice cultivation area of ~44 million hectares. The GBR models were trained per state, using the 5 km resolution, and then applied to the 500 m MODIS LAI original spatial resolution to predict rice yields at 500 m resolution. An out-of-sample validation of the developed model was conducted for Kharif seasons 2016 and 2017 at the district level. Figure 2 summarizes the stepwise workflow of the methodology. The input data preparation and model development are described in the following.

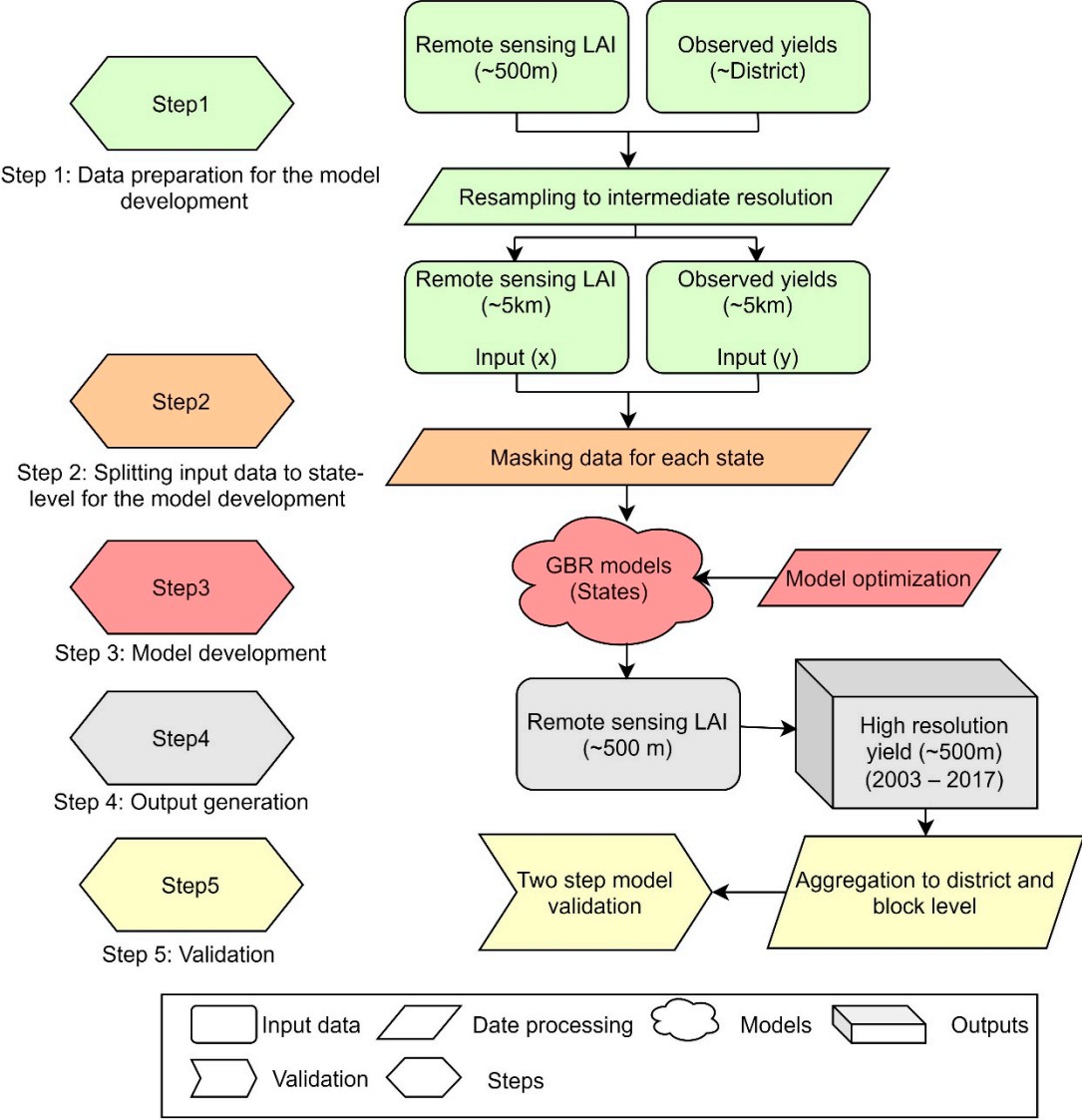

**Figure 2.** The methodology flowchart, including data pre-processing, model development, and validation.

### 2.3. Observed Yields

Observed yields were obtained from the Directorate of Economics and Statistics (DES), the mandated organization under the Ministry of Agriculture (MoA) for collecting and reporting agricultural production data in India. They release rice yield statistics at the district level; data for the years 2003–2017 were available. The yield statistics of major crops are attained by analyzing Crop Cutting Experiments (CCE) conducted under scientifically designed General Crop Estimation Surveys (GCES) conducted by a network of agricultural extension workers. However, these statistics often have data gaps; data availability is not possible immediately after the crop season due to its processing time (at least it takes another season). For filling missing data in the official yield statistics (DES), we also derived observed yields from Village Dynamics in South Asia (VDSA) survey created by the International Crops Research Institute for the Semi-Arid Tropics (ICRISAT) and the Bill & Melinda Gates Foundation (BMGF) [22]. VDSA provides an expanded and updated 19-states district-level database from 1966 to 2011. VDSA and DES data have a good agreement ($r = 0.97$) (Figure S1). We applied three filter steps in the official observed yield data to remove apparent outliers. The first step was to remove all yield values above 10,000 kg/ha as these were assumed not achievable at the district level in India. Second,

within a district time series, we removed those yield observations above or below the mean plus/minus two times the standard deviation. The third and final filter was to remove virtually constant time-series where the coefficient of variation (CV, defined as standard deviation over mean) was lower than 2% as these were deemed reporting errors. In addition, block-level (on average ten blocks per district) observed yields [23] were obtained for the state of Maharashtra (2003–2015) to validate the model with independent data further.

## 2.4. Leaf Area Index (LAI)—Input (x)

LAI is defined as the one-sided green leaf area per unit of the ground surface and is a dimensionless bio-physical parameter. In this study, LAI was derived from the Moderate Resolution Imaging Spectroradiometer (MODIS) at 500 m spatial resolution. For large-scale studies, MODIS is well apt due to its global coverage and for covering the whole growth period by getting multiple footprints for different dates [24,25]. We used the MCD15A2H Version 6, MODIS Level 4, LAI product for 2003 to 2017, at an 8-day temporal resolution [26]. We used 26-time steps (Julian Day 161 to 361), covering the full Kharif crop growth period (mid-May to end-November). An LAI value equal to zero represents bare soil, while high values (>5) account for a dense vegetation cover. We followed four steps to pre-process the data. The first step was to reproject the data from its default projection "sinusoidal" to "WGS 1984". Next, the data were filtered based on crop-specific LAI assessments [27,28] using the following rules: (a) The average LAI of the crop growth period is ≥0.5; (b) the maximum LAI of the crop growth period ≤5; (c) the standard deviation of the LAI in the crop growth period is ≥0.2. Third, a rice crop mask (see Section 2.5) was applied to the dataset, filtering out all LAI values outside the rice cropping areas. Finally, for the model training, LAI values were averaged to 5 km spatial resolution to match resampled observed rice yields.

## 2.5. Crop Mask

The crop mask for rice was taken from the Moderate Resolution Imaging Spectroradiometer (MODIS) multispectral rice classification [29]. This data is available for the 2000–2001 Kharif season at 500 m spatial resolution for South Asia under various irrigation categories. We used this map for all simulation years (2003–2017) because there have been no significant changes in crop area from 2003 to 2019 [30]. The spatial pattern of the MODIS rice mask is shown in Figure 1.

## 2.6. Quality Indicators for Model Optimization and Validation

Seven performance measures were used to optimize models and validate their outputs against observed yields. We used the coefficient of determination (R2) to find the best combination of parameters for each state. The subsequent in-sample and out-of-sample validation were assessed with Mean Absolute Error (MAE), Pearson's correlation coefficient (*r*), index of agreement (d) Nash–Sutcliffe model efficiency coefficient (NSE), and the Root Mean Square Error (RMSE). The MAE calculates the error between observed and simulated yields and is measured in the target variable units (t/ha). While the RMSE calculates how much error there is between observed and simulated yields, and the %RMSE calculates percentage error. The index of agreement (d) indicates the degree of model prediction error with values between 0 (no agreement) and 1 (perfect match). NSE is similar to $R^2$, measuring the explained variability between observed and simulated yields but also considering the level. It ranges from −Inf to 1. An NSE of 1 corresponds to a perfect match of simulated yields to the observed data (equations in Table S4).

## 2.7. Yield Modeling Approach

### 2.7.1. Gradient Boosted Regression (GBR) Trees

In this study, we applied the GBR approach to modeling rice yield from LAI satellite data. GBR consists of an ensemble of decision trees for classification or regression. A se-

quence of trees is created, where each tree in the sequence focuses on the previous tree's prediction residuals [31]. GBR can optimize different loss functions and provides several hyperparameter tuning options that make the function fit very flexible. It does not depend on variable pre-processing and can be implemented for categorical and numerical data. Another advantage of GBR is that it can handle missing data. Overfitting is accounted for by constructing simple trees in each iteration, which are more robustly extrapolated to independent data. GBR has been described in detail in the literature [32–37]. The GBR model parameters are discussed in Section 2.7.2. The python programming language and scikit-learn package [38,39] were used for this study.

### 2.7.2. Model Parameters

The parameters in GBR models can be divided into three categories, such as tree-specific parameters (impacts on decision trees), boosting parameters (impact on boosting operation, i.e., how the sequence of trees is generated), and miscellaneous parameters (impacts in overall functioning). In this study, we tuned the model with the tree-specific parameters "max_depth," "min_samples_split," and "min_samples_leaf," and boosting parameters "learning rate," and "n_estimators." The boosting parameters, "learning rate" and "n_estimators" were tuned simultaneously as both of the parameters highly depend on each other. The "n_estimator" specifies how many trees will be created in the model. The "learning rate" parameter affects each tree's contribution by the learning rate (i.e., a learning rate of 0.1 reduces the contribution of each tree by 90%). A lower learning rate means the model will allow the model to generalize better but requires a proportional increase in trees. The next parameter to be tuned and analyzed was the max depth parameter specifying how deep a tree can grow, and "min_samples_split" defines the minimum number of samples or observations that are needed in a node to be considered for splitting. Finally, "min_samples_leaf" defines minimum samples (or observations) required in a terminal node or leaf. As we split our data into states, we decided on the GBR parameters range based on the length of the observations per each state [36]. Combinations of the parameters shown in the supplemental information (Table S1) were tested over all the states.

### 2.7.3. Hyperparameter Tuning on the State Level

The parameter combinations shown in Table S1 were tested and tuned in the GBR models for each state. We used states as model units since the agricultural policies in India are designed and implemented by a complex system of institutions, and states have constitutional responsibility for many aspects of agriculture [40]. For choosing the best parameters for each state, the following steps were applied: (a) Initially, the input data were randomly split into training and test set with an 80:20 ratio, respectively. (b) Then, the multiple hyperparameter combinations (Table S1) were used to develop a model with the training data (80%) and validated against test data (remaining 20%), and estimated $R^2$. (c) Then the same multiple hyperparameter combinations were used for k-fold cross-validation. In k-fold cross-validation, we split our training data into three different subsets ($k = 3$). The multiple hyperparameter combinations were then used to develop GBR models in k-1 subsets to predict against the last fold to calculate $R^2$. We then averaged the three $R^2$ values of the out-of-sample folds. (d) Finally, the best combination was chosen out of all the combinations based on the highest sum of $R^2$ from the k-fold cross-validation (step c) and $R^2$ from step b. Where the $R^2$ was not able to separate models, we used the lowest of n_estimator, highest of the learning rate, lowest of max_depth, and maximum of min_samples_split, and min_samples_leaf, in that order, to avoid overfitting and reduce computational time. The final GBR model used for yield estimation at 500 m was derived from the full (100%) input data, using the combination of hyperparameters selected with the procedure above. After hyperparameter tuning over each state, the optimal input parameter combinations for model development and the relative importance of time-series LAI were determined. The flowchart (Figure 3) explains the stepwise methods in the best model optimization through hyperparameter tuning.

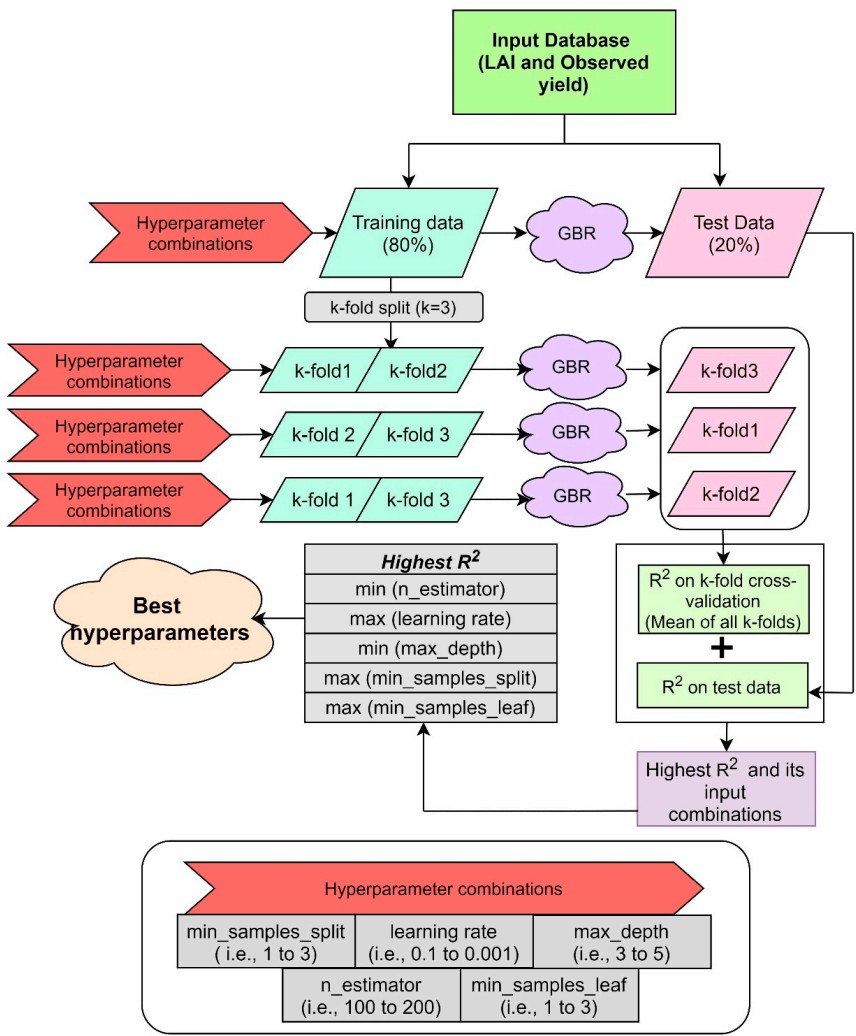

**Figure 3.** The step-wise hyperparameter selection for GBR models.

### 2.8. Two-Step Model Validation

After the optimization of the model input parameters, the best GBR model for each state was developed using 100% of input data on 5 km resolution and years 2003–2015 (~110,509 training points). These were then used to generate yield outputs at 500 m spatial resolution—the original MODIS LAI resolution—over the years 2003 to 2017. These 500 m spatial resolution estimated yields were aggregated at the district level and validated with the observed district-level yields. Next to comparison across all states and districts, comparisons were also made separately for different rice cropping area fractions (at least 5, 10, 15, 20, and 25%, respectively)—using the same GBR model—as management practices may diverge between different levels of rice prevalence. A temporal out-of-sample validation on district level was performed for the years 2016 and 2017, which were neither included in hyperparameter tuning nor model training data. For Maharashtra, estimates were also aggregated from 500 m to block-level for comparison with reported yields. Around 100 block-level yields were used for each year in this validation. We also compared the GBR model's performance with a multi-linear regression (Equation (S1)) to assess whether a simpler statistical method would yield similar levels of accuracy.

## 3. Results

### 3.1. Relative Importance of Crop Growth Period

The variable importance in a model is shown in relative percentages in Figure S2. Overall, the LAI values of Julian Days from 201 to 289 have higher importance than other

days' LAI values, which is approximately from germination/after transplantation to peak vegetation/grain filling stage for most of the states. Considering germination to peak vegetation stage is a decisive parameter of yield outcomes, some of the states (Uttarakhand and Rajasthan) have higher relative importance in a later stage than dates mentioned above, which could be due to late sowing. Few states like Andhra Pradesh, Odisha, and Jharkhand have higher relative importance in the early stage, which could be due to early sowing. In conclusion, irrespective of sowing dates, the vegetative phase is a crucial parameter of yield outcomes in all states.

### 3.2. Spatio-Temporal Performance of Downscaling Approach (2003–2015)

Rice yields simulated at 500 m spatial resolution (Figure S4) and re-aggregated to district level matched well with observed district-level yields (Figure 4). The quality indicators NSE, d, *r*, and R2, show a high model performance for the density-specific estimates, with all performance values being above 0.7 and RMSE (t/ha) values are between 0.36 to 0.51. Notably, the quality indicators are getting higher (>0.85) in districts with high rice density. The observed and simulated yields range is similar (between 0.5 and 5 t/ha), but there is a bias at the margins, where simulated yields show less pronounced spikes. State-wise aggregated yields also show a high correlation (d and *r* > 0.9) between observed and simulated yields (Figure S3, Table S2).

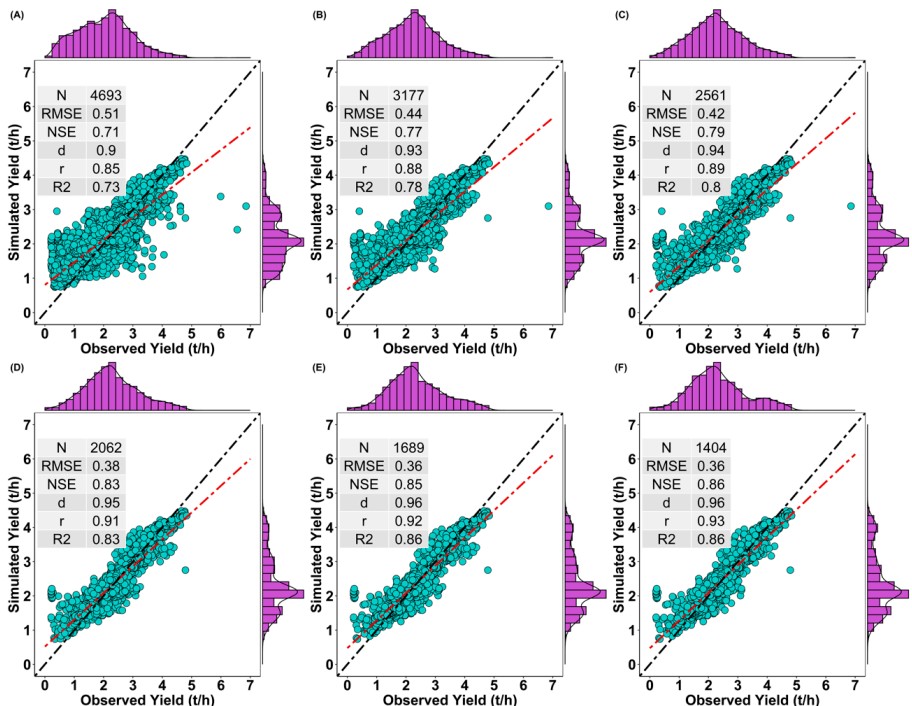

**Figure 4.** The comparison between observed and simulated yields for the years 2003–2015. The points in the plots represent districts. The red and black lines are representing the trend line and 1:1 line, respectively. The values for the indicators N, RMSE, NSE, d, *r* and R2 represent the number of observations, the Root Mean Square Error (t/ha), the Nash–Sutcliffe model efficiency coefficient, the index of agreement, Pearson's correlation coefficient, and Pearson's correlation coefficient of determination (R2). The histograms on the plot represent the yield distribution in the observed and simulated yields. Plots (**A–F**) are representing the comparison of observed and simulated yields for all districts, ≥5%, ≥10%, ≥15%, ≥20%, and ≥25% crop area density districts, respectively. These ranges are set with an open interval, i.e., observations overlap such that the points shown in (**F**) are contained in all other panels (**A–E**), and so on.

### 3.3. Inter-Annual Yield Variability

Time series of observed and simulated yields at the state level agree well for most states (Figure 5). The ability to capture the annual yield variability is important for downscaling model results. The simulations in water-limited states with predominant rainfed agriculture (like Chhattisgarh, Jharkhand, Odissa) have performed well with a correlation coefficient ($r$) of >0.9. The model was able to capture all of the peaks and downs in these states. States with high irrigation share (Andhra Pradesh, Haryana, Karnataka, Punjab, Telangana, Uttara Pradesh, and West Bengal) had a simulation performance with $r \geq 0.78$. States with relatively marginal rice growth areas (Maharashtra, Madhya Pradesh, Jammu, and Kashmir) also show a strong correlation ($r \geq 0.85$) between observed and simulated yields. However, in three states (Bihar, Himachal Pradesh, Rajasthan, and Uttarakhand), the model does not capture the observed variability.

**Table 1.** The values for the agreement indicators between state-wise observed and simulated yields (Figure 5), Observations, $r$, R2, MAE, and RMSE represent the number of observations, Pearson's correlation coefficient ($r$), the Pearson's correlation coefficient of determination (R2), Mean Absolute Error (t/ha), and the Root Mean Square Error (t/ha), respectively.

| State | Observations | $r$ | R2 | MAE (t/ha) | RMSE (t/ha) |
|---|---|---|---|---|---|
| Andhra Pradesh | 13 | 0.94 | 0.88 | 0.13 | 0.15 |
| Bihar | 9 | 0.28 | 0.08 | 0.35 | 0.55 |
| Chhattisgarh | 13 | 0.99 | 0.97 | 0.11 | 0.12 |
| Gujarat | 13 | 0.54 | 0.29 | 0.21 | 0.24 |
| Haryana | 10 | 0.82 | 0.67 | 0.11 | 0.13 |
| Himachal Pradesh | 6 | −0.19 | 0.04 | 0.16 | 0.21 |
| Jammu & Kashmir | 5 | 0.96 | 0.93 | 0.23 | 0.26 |
| Jharkhand | 7 | 0.99 | 0.97 | 0.27 | 0.31 |
| Karnataka | 13 | 0.79 | 0.62 | 0.13 | 0.16 |
| Madhya Pradesh | 11 | 0.93 | 0.86 | 0.18 | 0.2 |
| Maharashtra | 13 | 0.85 | 0.72 | 0.24 | 0.26 |
| Odisha | 9 | 0.94 | 0.88 | 0.04 | 0.04 |
| Punjab | 13 | 0.78 | 0.6 | 0.12 | 0.15 |
| Rajasthan | 13 | −0.16 | 0.03 | 0.21 | 0.26 |
| Telangana | 13 | 0.91 | 0.83 | 0.12 | 0.16 |
| Uttar Pradesh | 13 | 0.96 | 0.92 | 0.14 | 0.15 |
| Uttarakhand | 13 | 0.13 | 0.02 | 0.3 | 0.31 |
| West Bengal | 9 | 0.83 | 0.69 | 0.08 | 0.09 |

### 3.4. District-Wise Model Performance

On the district level, observed and simulated yields agree well in large parts of the 18 studied states (Figure 6). The correlation coefficient and the index of the agreement, the Root Mean Square Error, the correlation coefficient determination (R2), and the % Root Mean Square Error are associated with crop density. Districts with a rice area fraction of at least 1–5% obtained a better agreement than districts with smaller growing areas, where the yield modeling approach is reliable. Overall, 63% of districts (out of 363) obtained a correlation coefficient and index of agreement larger than 0.5. Districts with at least 5% rice crop density showed higher performance, having 75% of districts (out of 247) with correlation coefficient and index of agreement larger than 0.5.

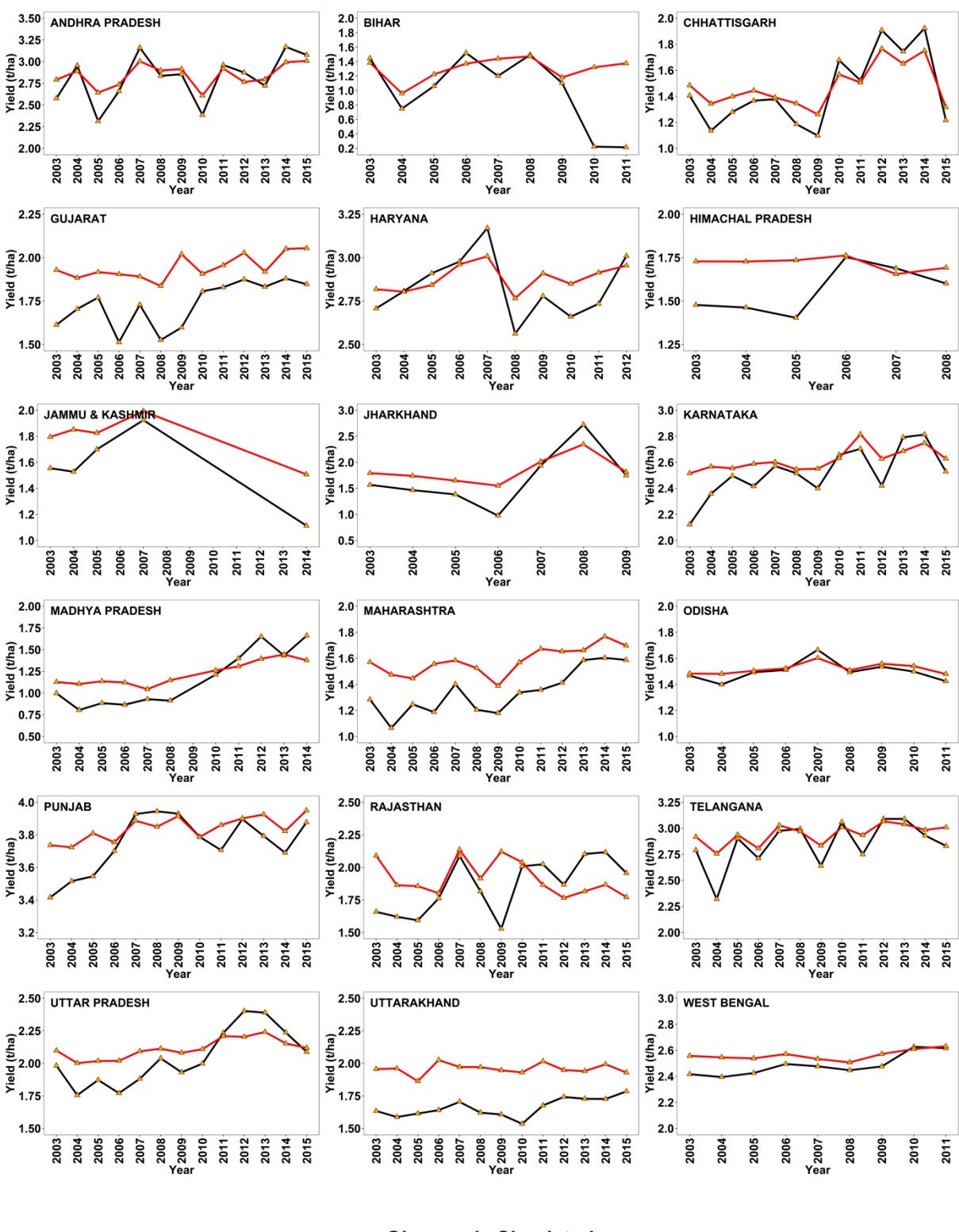

**Figure 5.** Time series information of state aggregated observed and simulated yields. Table 1 presents the agreement indicators between Observed and Simulated yields at state wise.

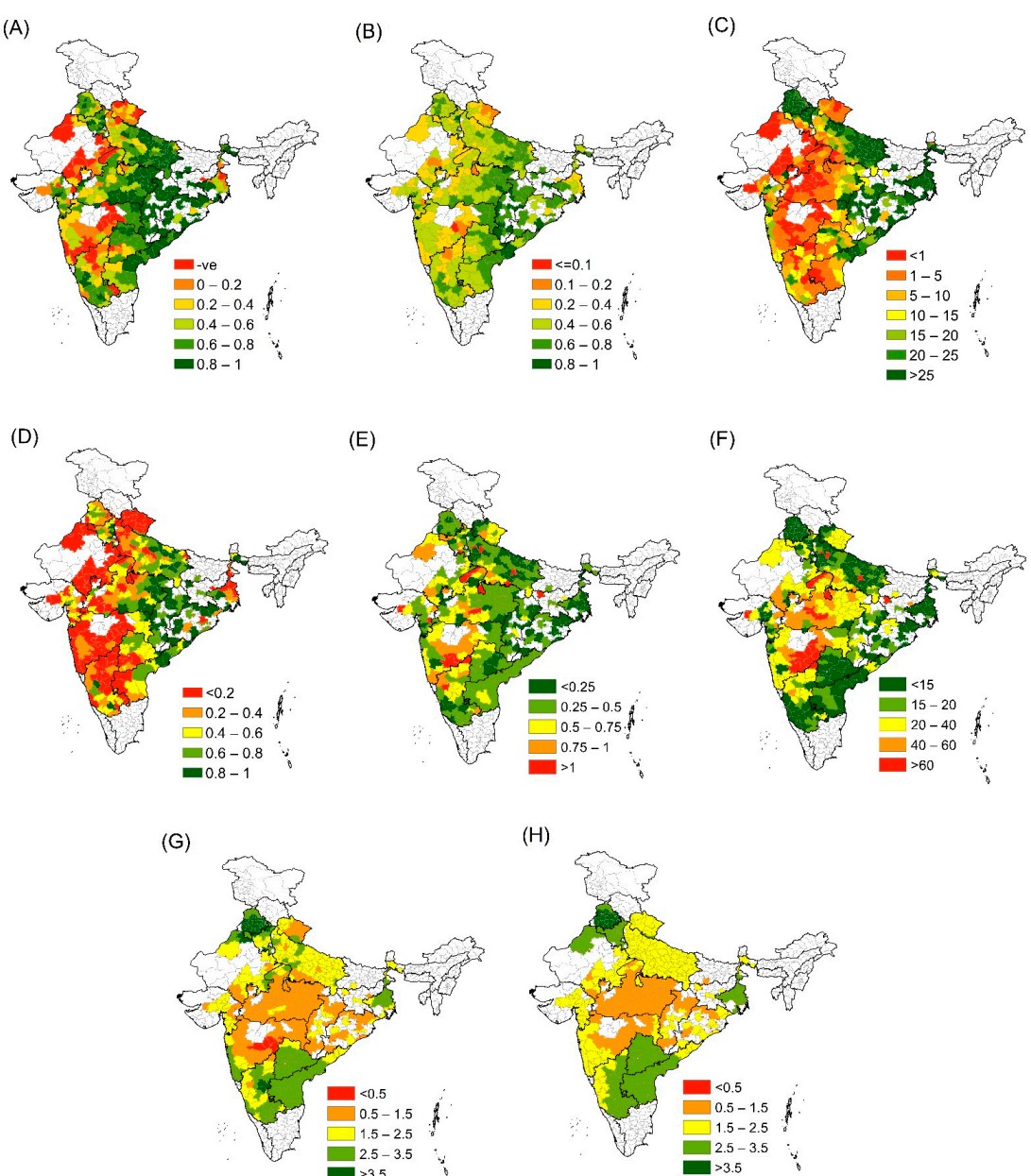

**Figure 6.** Spatial distribution maps for (**A**) district-level correlation coefficient (*r*), (**B**) the index of agreement (d), (**C**) the rice density (area %), (**D**) the Pearson's correlation coefficient of determination (R2), (**E**) the Root Mean Square Error (t/ha), (**F**) the Percentage Root Mean Square Error (%), (**G**) long-term (2003–2015) average observed yields, and (H**)** long-term (2003–2015) average simulated yields.

### *3.5. Block-Level Validation*

We also validated the GBR-derived 500 m yield estimates with reported Maharashtra block-level yields from 2003 to 2015. Model performance is lower at the block level and shows a bias at both margins, underestimating high yields (above 2.5 t/ha) and overestimating low ones (below 1 t/ha), but capturing the slope (*r* = 0.7) (Figure 7, Table S3).

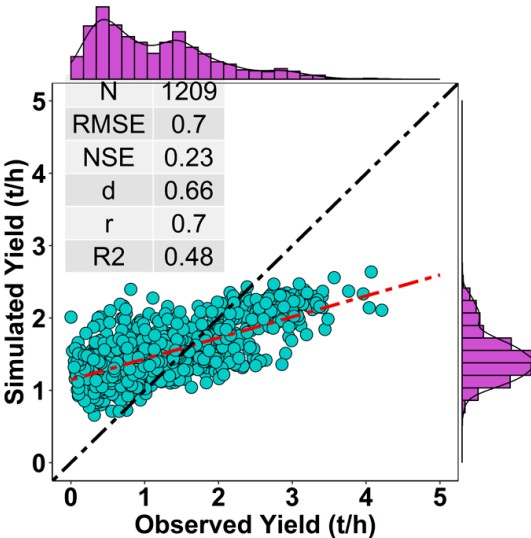

**Figure 7.** The Spatio-temporal correlation between block-level observed and simulation yields for Maharashtra from 2003 to 2015. The points in the plots represent districts. The red and black lines are representing the trend line and 1:1 line, respectively. The values for the indicators N, RMSE, NSE, d, *r*, and R2 represent the number of observations, the Root Mean Square Error (t/ha), the Nash–Sutcliffe model efficiency coefficient, the index of agreement, Pearson's correlation coefficient, and Pearson's correlation coefficient of determination (R2). The histograms on the plot representing the yield distribution in the observed and simulated yields.

### 3.6. Out of Sample Validation

We used the 2016 and 2017 district-wise observed yields to validate the models trained on 2003–2015 independently. The results (Figure 8) indicate that the model estimation can robustly be extrapolated to independent data in adjacent years. Notably, though, performance in 2017 decreases slightly when compared to 2016.

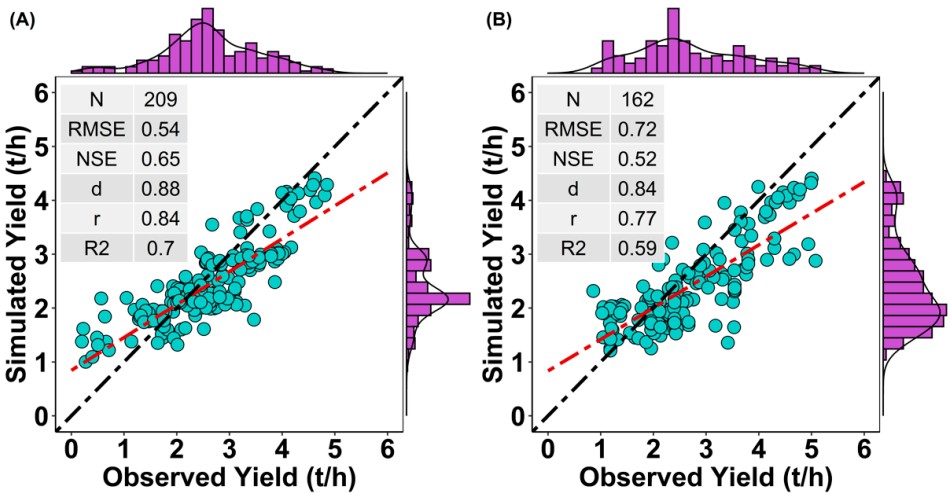

**Figure 8.** The correlation between observed and simulated yields for the out of sample validation years. The points in the plots represent districts. The red and black lines are representing the trend line and 1:1 line, respectively. The values for the indicators N, RMSE, NSE, d, *r* and R2 represent the number of observations, the Root Mean Square Error (t/ha), the Nash–Sutcliffe model efficiency coefficient, the index of agreement, Pearson's correlation coefficient, and Pearson's correlation coefficient of determination ($R^2$). The histograms on the plot representing the yield distribution in the observed and simulated yields. Plots (**A**,**B**) are representing the comparison of observed and simulated yields for the years 2016 and 2017, respectively.

## 4. Discussion

### 4.1. Data and Model Quality

In terms of availability of observed yields (training data) for regional-level yield assessment models, India has sufficient observations (district-level) except East Central India (Chhatisgarh, Orrisa, Jharkhand, and Bihar) and Central India (Madhya Pradesh and Maharashtra). However, for field-level yield assessments, ground-level observations would be appreciable. Nevertheless, this downscaling approach could be a bridge to set regional-level observations for large-scale moderate-resolution (500 m) outcomes. Considering remote sensing data, global maps of LAI derived from the MODIS reflectance data are an essential resource in global change studies, but errors/outliers in the product must be characterized and well understood. Some of the LAI images were not usable due to high standard deviations of band reflectance, probably due to the combination of images with different atmospheric conditions and cloud cover that could not be detected [41]. Nevertheless, the utilization of crop masks/filtering techniques (Section 2.5) and multiple time-steps in the model development could be a partial solution for atmospheric disturbance such as cloud cover and non-selective scattering due to water droplets and large dust particles. In the matter of spatial resolution, we used 500 m spatial resolution even with the availability of higher resolution data such as Sentinel (~10 m) and Landsat (~30 m) for the following reasons: (a)There is a strong limitation of observed data to train/validate models at higher resolution. (b)A similarly high resolved crop classification (land use map) at the national scale is not available. (c)Short historical coverage of Sentinel (only from 2015) and limitation of cloud-free images in Landsat due to longer revisit times (30 m).

Considering the chosen algorithmic approach, recent studies reported [42–44] that GBR is one of the best predictive modeling approaches in deriving crop and soil inventories using vegetation, soil, and terrain information. This is because GBR can optimize on different loss functions and provides several hyperparameter tuning options that make the function fit very flexible and requires no data pre-processing [31,45]. Also, boosted trees can handle missing values/outliers by creating surrogate splits that behave like the split containing missing observations [46], which helps for missing LAI data. However, boosted models are sensitive to the number of observations in the predictive modeling approach, and it requires more training data than other linear and ML approaches [47]. Also, boosted trees are very susceptible to overfitting, as the boosting algorithm focuses on smaller subsets of the data in later iterations [48]. Efficient implementations of gradient tree boosting have a large number of hyperparameters. This renders the optimal tuning of the model challenging due to the high dimensional configuration space [49]. Therefore, to avoid over-fitting, reduce computational time, and create better predictive models, we chose the approach of multiple layers of out-of-sample testing for this study. Many machine approaches such as Random Forest, Support Vector Machine, Neural Network, etc., are suitable for classification and clustering. In terms of the predictive modeling approach, GBR is the most suitable approach compared to other machine learning approaches [50]. Additionally, we also developed linear models for each state and compared them with GBR models. Linear and GBR models were developed with 80% input data and applied on test data (remaining 20%) as predictive models. The R2 values of the GBR models with the test data were an average of 220% higher than the linear models (Figure S5 and Table S5). This shows that there is a skill in GBR models compared to the linear models.

### 4.2. Performance of the Downscaling Approach in Yield Estimation

The model's ability to capture the interannual variability for the water-limited states indicates that it can play a vital role in agronomic decision support such as crop insurance applications. In eastern India, studies have reported that enormous losses of rice production occur from drought and high runoff losses due to storms that cause much soil and nutrient erosion [51,52]. Our study approach could be beneficial in such conditions to compensate for the yield loss to farmers as this method can provide yield data at a near-real-time and 500 m spatial resolution. In addition, the model's ability to capture variability in

irrigated states was remarkable. It can explicate that the crop yield variability can also be strongly influenced by other factors than water availability [53] and that the model—though agnostic about the mechanistic causes—can capture non-water impacts on yields. In Bihar, for 2010 and 2011, there was only one observation (district) per year available for the whole state, which could be the reason for not matching the simulation. Some of the northern states, such as Rajasthan and Himachal Pradesh's poor model performance, can be explained by the lowest crop area (on average, 2.1%/district) in comparison to other states. The simulation was not able to reproduce very low (<0.5 t/ha) and high yields (> 5 t/ha), likely because there were not sufficiently many observations to train the model [54]. Therefore, the model's results for states where rice is not dominant should be used with caution. In terms of model applicability in the future—which we assessed with a temporal out-of-sample validation for two subsequent years—models must be trained with the available latest observations from last year's harvest. This ensures a better consideration of the role of farming technologies in yield improvement. An annually updated crop mask could also increase accuracy (not tested here).

The model's ability was directly proportional to the crop density (i.e., higher density crop districts have produced good accuracy). The higher crop density leads to higher crop cutting samples, which are highly reliable, mainly crop area density with at least 1–5%. Some studies [55–59] used linear/machine learning approaches to predict rice yields using various remote sensing indices and microwave remote sensing. However, these studies are restricted to a limited spatial extent. Two previous studies [60,61] used biophysical models to estimate crop yields at coarser resolution (~5 km & ~50 km) over India, which is not sufficient for village-level applications. Comparing block-level yields, the disagreement between observed and simulated yields is higher than for districts since we trained our model with district-level yields, containing a lower range of variation likely due to aggregation effects. We conclude that we can use our yield simulations at 500 m spatial resolution (~approximately 25 ha) for regional agricultural applications as our model performed well at the district and acceptable at the block level. Field level assessments in the Indian agricultural landscape needed higher resolution images such as IKONOS and SPOT due to their crop heterogeneity. However, these 500 m resolution outcomes could be helpful for agricultural applications at the village scale, such as planning market strategies and crop insurance. The current model approach is required dense ground truth observations for specific cropping conditions in model training, especially in extreme higher and lower yields. This helps to attain its best accuracy for field-level applications, circumventing further research following [62].

### 4.3. Recommendations for Future Research

In this study, we used only one variable (LAI) as exogenous input and used only one ML method. Therefore, we see our study as an initial step in large-scale near-real-time yield estimation and recommend using multiple data sources such as vegetation indices, soil/terrain information, hydrological information, and socioeconomic parameters for better model developments and assessing the importance of factors in agriculture. Examples of sources for high-resolution images are Unmanned Aerial Vehicles (UAV) images, IKONOS, SPOT, and WorldView satellites. New methods and innovative ways of using ML approaches to upscaling these images will create many opportunities in the developing world with a high-dimensional configuration space.

### 5. Conclusions

We have presented a downscaling approach to estimate rice yields in India on 500 m spatial resolution. Our approach achieves high accuracy for most districts and states in within-sample and out-of-sample validation. The novel method integrates remote sensing data with a machine learning technique that could potentially be applied for timely and reliable yield estimation. This study demonstrated the advantage of machine learning methods and substantial spatial coverage of remote sensing data in crop yield estimation.



This study also assessed the relationship between observed yields and remote sensing data using Gradient Boosted Regression (GBR), a machine learning technique in coarser resolution, and then applied in higher resolution data. The results are promising that this downscaling approach to estimate yield is appropriate for rice in India. In summary, our results show that GBR can be an efficient downscaling approach for simulating crop yield at a national scale. Our approach could play a vital role in estimating accurate crop yields and agricultural externalities under a wide range of environmental and management conditions and for loss determination of insurance schemes. Besides, this approach also does not rely on high-resolution ground data and can thus potentially be deployed for agricultural assessments in other parts of the world.

**Supplementary Materials:** The following are available online at https://www.mdpi.com/article/10.3390/rs13122379/s1. Table S1: Parameters tested in the hyperparameter tuning to find the best model from GBR decision trees for all the states, Table S2: The state-wise aggregated yield (2003 to 2015) of observed and simulated, Table S3: The temporal validation between block-level observed and simulated yields. N, d, and r represent observations, the index of agreement, and the correlation coefficient, respectively, Table S4: The parameters used for analyzing the assessment of the goodness index, Table S5: Comparision between GBR and Linear model's (trained with 80% of data) accuracy on test data (validated with 20% of data) for each state with the difference increased in GBR from linear models, Equation S1: Multiple linear regression model, Figure S1: Comparison between two different observed yields, VDSA and DES, Figure S2: State-wise relative importance (%) of input parameters (time-series LAI values) in GBR, Figure S3: The comparison between state aggregated over the model development years (2003–2015) of observed and simulated yields. The points in the plots are representing states. The red and black lines are representing the trend line and 1:1 line, respectively. The values for the indicators N, NSE, d, and r are representing the number of observations, Nash–Sutcliffe model efficiency coefficient, the index of agreement and the Pearson's correlation coefficient, respectively. The histograms on the plot representing the yield distribution in the observed and simulated yields, Figure S4: Spatial distribution of simulated yields from the year 2003 to 2017, Figure S5: Comparision between GBR and Linear model's (trained with 80% of data) accuracy on test data (validated with 20% of data) for each state.

**Author Contributions:** Conceptualization, P.A.; methodology, P.A., A.C., B.S. and C.G.; model development, P.A.; validation, P.A. and A.C.; investigation, B.S. and C.G.; writing—original draft preparation, P.A. and A.C.; writing—review and editing, B.S. and C.G.; visualization, P.A.; supervision, B.S. and C.G. All authors have read and agreed to the published version of the manuscript.

**Funding:** The East Africa Peru India Climate Capacities (EPICC) project funded this research with the International Climate Initiative (IKI) funded by the German Federal Ministry for the Environment and Nature Conservation, and Nuclear Safety (BMU).

**Institutional Review Board Statement:** Not applicable.

**Informed Consent Statement:** Not applicable.

**Data Availability Statement:** The data presented in this study are available on request from the corresponding author. The data are not publicly available due to conditional pre-processing, and it is voluminous.

**Acknowledgments:** We are grateful to Moderate Resolution Imaging Spectroradiometer (MODIS) team, Directorate of Economics and Statistics (DES), Village Dynamics in South Asia (VDSA), and International Rice Research Institute (IRRI) for making their data available for this analysis. We want to thank East Africa Peru India Climate Capacities (EPICC) program for funding this research within the International Climate Initiative (IKI), funded by the German Federal Ministry for the Environment and Nature Conservation and Nuclear Safety (BMU).

**Conflicts of Interest:** The authors declare no conflict of interest.

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
