# Peer review of "Remote Sensing Based Yield Estimation of Rice (Oryza Sativa L.) Using Gradient Boosted Regression in India"

_remotesensing, doi:10.3390/rs13122379_

Round 1

Reviewer 1 Report

Dear authors,

The study called “Remote Sensing Based Yield Estimation of Rice (Oryza Sativa 2 L.) Using Gradient Boosted Regression In India” discusses the potential to estimate rice yield in India using MODIS-derived LAI and ML in the majority of rice-producing Indian regions. Yield estimation is a central topic for global sustainability and farmers socioeconomic well-being and thus deserves attention, especially in a country highly dependent on agriculture. Moreover, ML actual performances on yield estimations is of central interest for the scientific community as it aims to provide tools for an improved agricultural planning and management. Nonetheless, in the current form the article is not suitable for publication in Remote Sensing (MDPI) and an extra effort is needed in improving the manuscript.

In general terms the article is well written, however there is a lack of clarity on the objectives and in key methodological approaches which should be modified. I believe there is a misuse of the term high-resolution yield estimates (line 241, 347). Currently with the use of openly accessible Sentinel-2 imagery, for instance, and GPS combine harvesters, there are several articles that have matched multi-level sensing platforms to obtain 10 m resolution yield maps.

Hunt, Merryn L., et al. "High resolution wheat yield mapping using Sentinel-2." Remote Sensing of Environment 233 (2019): 111410.

Kayad, Ahmed, et al. "Monitoring within-field variability of corn yield using Sentinel-2 and machine learning techniques." Remote Sensing 11.23 (2019): 2873.

It is understandable that the estimation capacities depend highly on the country/region or the study scope, nonetheless in more complex agricultural landscapes (such as Mali) relatively high-resolution yield maps have also been achieved.

Lambert, Marie-Julie, et al. "Estimating smallholder crops production at village level from Sentinel-2 time series in Mali's cotton belt." Remote Sensing of Environment 216 (2018): 647-657.

Therefore, the use of the word high-resolution should be revised. In addition to this, the actual 500 m resolution of the model is questionable. How can a yield estimation model be generated with 5 km yield feeding data? What is the sense to resample the model to 500 m (Figure 2) when there is not validating data at this resolution level? In this sense, I would suggest to re-structure the manuscript focusing on 5km resolution if that is the data available for validation. As far as I understand, the "out of sample validation" section (line 328) is also at this level, namely at 5 km.

Another general commend is on the use of only one specific ML approach and the statistical summaries chosen. I would suggest adding at least a baseline model to compare to the chosen ML approach. In this sense a simple linear regression, for instance, should be added in the study. There is not clear evidence that the model presented in the manuscript is better than a linear regression or any other modelling approach. Thus, I suggest adding a comparison part with an “a priori” less efficient model. Regarding the statistical summaries, the benefits of using NSR in contrast with R2, or MSA in contrast with RMSE are not clear. I would suggest providing R2 and RMSE values instead of the used ones as they are more understandable. Moreover, RMSE% could be added for an improved clarity. The average of yield per district is not presented in the article, therefore the actual error is difficult to understand.

Please, find in the following lines several comments for each section of the manuscript.

Introduction

The introduction is very well written, nonetheless I would suggest writing the specific objectives of the study in the last paragraph as well as the modifications suggested. It is important to introduce the steps of the analysis (line 68), but what is aimed should be clear. In this case is obtaining accurate rice yield estimation models at regional level the aim? The objective is not clear, do you aim to test a specific ML approach to estimate yields? Do you aim to generate a replicable model to be used by institutions and insurance companies? Does FASAL have not used ML yet and those are the first advancements?

Revision points highlights:

-The aims of the study should be clarified in the introduction.

Materials and Methods

Related revision points highlights:

-I have not found the number of samples you are working with, the number of samples should be stated (2,000: 3,000…) for the case of 500m LAI and district/level rice yields. (In line 364 something is writing in this sense).

-Line 68: what is the resolution of grain yields? Field, local, district level? If the resolution is district level (5 km) why do you focus on 500 m when there is not validation data.

-Lines 169-166: I would suggest to better use R2 and RMSE and the %RMSE is not providing yield averages per district.

Results

The figures are very well-presented, they should only be modified accordingly to the suggested revision points.

Discussion and conclusions

Both should be revised according to the suggested modifications.

I hope this message finds you well and can contribute to improved the manuscript.

Best wishes

Author Response

Dear Reviewer,

We have attached the response to the reviews in the attached document. Please find it. 

Thanks and regards

Author and Co-authors

Reviewer 2 Report

The paper is done according to the instructions given in the journal guidelines. Organization of paper with sections (Introduction, Materials and Methods, Results, Discussion and Conclusions) is adequate with.

Supplementary Materials is well done and material is ordered in a way that is logical, clear, and easy to follow.

Author cited sources adequately and appropriately, and all the citations in the text are listed in the References section. English language and style are fine.

Research data from the study area and experiment results are presented and visualized in clear way, which all makes paper readable. Paper is well structured and well written. 

This paper presents study on estimation rice yields in India with methodology that integrates remote sensing data with a machine learning technique

Authors succeed to show that Gradient Boosted Regression (GBR) can be an efficient downscaling approach for simulating crop yield. 
There is no any novel methods or techniques, but it is very well modelling. The study used one variable as input and one method.

Author Response

We thank the reviewer very much for the positive reception of our manuscript.

Thanks and regards

Author and Co-authors

Reviewer 3 Report

This is an ambitious work to estimate the yield of rice crops in the whole country (India) using multispectral MODIS images.

The introduction presents previous studies in India to forecast crop yields using Remote Sensing techniques. The authors indicate that the novelty of their work is mostly based on the use of AI and the availability of a time series of actual crop yield data to apply Gradient Boosted Regression to forecast rice yield.

The authors used the MODIS products of LAI calculations and the reports of rice crops yields for the period from 2003 to 2015 to perform the regression calculation and the LAI values from 2016 and 2017 to validate the results. The authors performed a good filtering process of the reported yield data to remove reporting errors.

In Figure 3. The plot represent: all 281 districts, ≥5%, ≥10%, ≥15%, ≥20%, and ≥25% crop area density districts, is it just greater than or is it 5% <crop density<10, i.e. is it a limited interval or an open interval, if it is open then the results would overlap. The authors should explain with more detail this result that would imply some restrictions to the application of their model, as it would produce good results only for areas with high crop density. It would be important to define a limit for its application.

The authors state: “The results (Figure 7) indicate that the model estima-329 tion can robustly be extrapolated to independent data in adjacent years”; however, this figure presents an important decrease in the accuracy of the simulated yield, the authors must explain this and provide a solution. Apparently the simulation is only valid for the specific conditions of the construction of the model and for years with different conditions its applicability is reduced. Is it possible that the model application will improve after many years of training with measured data?

It would be important to provide a sensibility estimation of the model for the LAI calculation.

The results, as presented in the available information, seem to yield interesting results with acceptable correlation between LAI calculated with satellite images and the measured data on rice crops yield, mostly considering the wide range of conditions in the study area.

Author Response

We thank the reviewer very much for taking the time to inspect our manuscript, its positive reception, and the helpful suggestions where to improve it. We have attached a response to the reviewer in the attached document. Please find it. 

Thanks and regard

Author and Co-authors

Reviewer 4 Report

The paper Remote Sensing Based Yield Estimation of Rice (Oryza Sativa 2
L.) Using Gradient Boosted Regression In India is well structured and well written. However, there are some parts that needs to be improved before its consideration for publication.

The paper uses GBR for Rice estimation in India using low-resolution remote sensing imagery. It should be mentioned what amount of rice can be estimated with this resolution and it should be discussed the impact of the resolution on the results of the study.

The introduction must be improved. The hypothesis of the paper needs to be clearly stated.

The model is well explained.

The results must be discussed with similar studies, did the model preformed better, was are the advantages and disadvantages etc.

Author Response

We thank the reviewer for taking the time to review our study and providing pertinent comments for improvement. We have attached the response to the reviewer in the attached document. Please find it.

Thanks and regards

Author and co-authors

Reviewer 5 Report

This manuscript reports a research to solely develop the Gradient Boosted Regression (GBR) trees approach to estimation of rice yield in India using MODIS LAI remote sensing product. To facilitate the model and characterization of the observed data variation the process switched between scales and the results show good agreement with official statistics and observations. The questions remain as

  1. There are so many ways to model and estimate yield from LAI. This research is only on GBR. So, what are the characteristics for GBR to uniquely fit this study? Based on the description of the current manuscript it is not convincing that GBR is unique or best for this study.
  2. What is the model structure of the GBR trees between LAI and yield? What is the physical meaning of the model structure?

Author Response

We thank the reviewer for the critical inspection of our manuscript and the helpful suggestions for improvement. We have attached the response to the reviewer in the attached document. Please find it.

Thanks and regards

Author and Co-authors

Reviewer 6 Report

The authors have done a phenomenal job with this manuscript. It was a pleasant and interesting read. The authors have provided a timely description of a machine learning technique to remote sensing data for the specific purpose of estimating yields in areas where high-resolution (and high-coverage) remote sensing data may not be available. While the tools used are in and of themselves not entirely new, their application in this manner and proof of concept are incredibly important in the discipline. I also appreciate the authors' discussion of limitations and needed areas of future improvement (e.g. cloud cover and image variability), as well as the authors' meticulous description of the methodology. The only suggestions I have are minor grammatical edits. 

Lines 44-45: i.e. should be e.g., and "The" should not e capitalized. 

Line 53: estimating should be estimate

Line 79: there is a missing "and". 

Lines 166-169: The description of Pearson's correlation coefficient is not needed.

Table 1 feels a bit unnecessary. Perhaps move to supplemental. 

Fantastic job. 

Author Response

We thank the reviewer for the inspection of our manuscript and the positive reception. We have revised all minor errors mentioned below.

Thanks and regards

Author and co-authors

Round 2

Reviewer 4 Report

The authors have improved the paper and now it is suitable for acceptance.

Reviewer 5 Report

The concerns have been well addressed. Thanks!